**Data Availability Statement:** The data has been provided in the supporting information files.

# Dietary adherence and program attrition during a severely energy-restricted diet among people with complex class III obesity: A qualitative exploration

**Gabrielle Maston**[1,2]*, **Janet Franklin**[2], **Samantha Hocking**[1,2,3], **Jessica Swinbourne**[2], **Alice Gibson**[4], **Elisa Manson**[2], **Amanda Sainsbury**[5], **Tania Markovic**[1,2,3]

1 The Boden Collaboration for Obesity, Nutrition, Exercise & Eating Disorders, Charles Perkins Centre, The University of Sydney, Camperdown, NSW, Australia, 2 Metabolism & Obesity Services, Royal Prince Alfred Hospital, Camperdown, NSW, Australia, 3 Sydney School of Medicine (Central Clinical School), Faculty of Medicine and Health, The University of Sydney, Camperdown, NSW, Australia, 4 Menzies Centre for Health Policy, School of Public Health, Faculty of Medicine and Health, The University of Sydney, Camperdown, NSW, Australia, 5 School of Human Sciences, Faculty of Science, The University of Western Australia, Crawley, WA, Australia

* gmas2269@uni.sydney.edu.au

## Abstract

Meal replacement Severely Energy-Restricted Diets (SERDs) produce $\geq$ 10% loss of body mass when followed for 6 weeks or longer in people with class III obesity (BMI $\geq$ 40 kg/m²). The efficacy of SERDs continues to be questioned by healthcare professionals, with concerns about poor dietary adherence. This study explored facilitators and barriers to dietary adherence and program attrition among people with class III obesity who had attempted or completed a SERD in a specialised weight loss clinic. Participants who commenced a SERD between January 2016 to May 2018 were invited to participate. Semi-structured in-depth interviews were conducted from September to October 2018 with 20 participants (12 women and 8 men). Weight change and recounted events were validated using the participants' medical records. Data were analysed by thematic analysis using line-by-line inductive coding. The mean age ± SD of participants was 51.2 ± 11.3 years, with mean ± SD BMI at baseline 63.7 ± 12.6 kg/m². Five themes emerged from participants' recounts that were perceived to facilitate dietary adherence: (1.1) SERD program group counselling and psychoeducation sessions, (1.2) emotionally supportive clinical staff and social networks that accommodated and championed change in dietary behaviours, (1.3) awareness of eating behaviours and the relationship between these and progression of disease, (1.4) a resilient mindset, and (1.5) dietary simplicity, planning and self-monitoring. There were five themes on factors perceived to be barriers to adherence, namely: (2.1) product unpalatability, (2.2) unrealistic weight loss expectations, (2.3) poor program accessibility, (2.4) unforeseeable circumstances and (2.5) externalised weight-related stigma. This study highlights opportunities where SERD programs can be optimised to facilitate dietary adherence and reduce barriers, thus potentially improving weight loss outcomes with such programs. Prior to the commencement of a SERD program, healthcare professionals facilitating such programs

**Funding:** AS was supported by a Senior Research Fellowship (APP1135897) from the National Health and Medical Research Council (NHMRC) of Australia. The funders had no role in study design, data collection and analysis, decision to publish, or preparation of the manuscript.

**Competing interests:** AS is the author of The Don't Go Hungry Diet (Bantam, Australia and New Zealand, 2007) and Don't Go Hungry For Life (Bantam, Australia and New Zealand, 2011). She has also received payment from Eli Lilly, the Pharmacy Guild of Australia, Novo Nordisk, the Dietitians Association of Australia, Shoalhaven Family Medical Centres, the Pharmaceutical Society of Australia, and Metagenics, for presentation at conferences, and served on the Nestlé Health Science Optifast® VLCD™ Advisory Board from 2016-2018. TM has received payments from Novo Nordisk for seminar presentations at conferences. She has served on the Egg Nutrition Council for Australian Eggs since 2004, Nestle Health Science VLCD Advisory Board since 2010 and Eli Lilly New South Wales Advisory Board since 2020. JF has received payments from Novo Nordisk, Illy, Boehringer Ingelheim and Nestle Health Science for seminar presentations. She served on the Nestle Health Science VLCD Advisory Board between 2009-2013. AAG has received payment from the Pharmacy Guild of Australia and from Nestlé Health Science for oral presentations at conferences. SH has received honoraria from Novo Nordisk, Inova, Sanofi, Eli Lilly, Boehringer Ingelheim and Astra Zeneca for seminar presentations. She has received research funding from Novo Nordisk. GM is the author of The Perfect Juice (New Holland, 2016) and received payment from Novo Nordisk for seminar presentations. JS reports personal fees from Novo Nordisk (Australia), Novo Nordisk Global (Denmark), Coloplast (Australia) and Coloplast Global (Denmark), DesignPsykologi (Denmark); she is on the advisory panel of SmartShape Centre for Weight Management outside the submitted work. Our declared competing interests do not alter our adherence to PLOS ONE policies on sharing data and materials.

could benefit from reviewing participants to identify common barriers. This includes identifying the presence of product palatability issues, unrealistic weight loss expectations, socioeconomic disadvantage, and behaviour impacting experiences of externalised weight-related stigma.

## Introduction

Class III obesity, defined by the World Health Organisation (WHO) as a body mass index (BMI) $\geq$ 40 kg/m$^2$ [1], poses a significant health risk to the individual. While excess fat mass increases the risk of conditions such as type 2 diabetes [2, 3], cardiovascular disease [4], obstructive sleep apnoea [5], kidney disease [6] and musculoskeletal problems leading to physical disability [7, 8], a 10% reduction in body weight can reduce the risk of obesity-related comorbid conditions [9–12]. People with obesity-related health complications in addition to class III obesity (i.e., people with complex class III obesity) are a unique group of people that are not well researched [13]. Often, subjects with uncomplicated overweight and obesity are recruited for clinical weight loss trials as mixed cohorts, alongside people with complex class III obesity [14–16]. As such, there is little available data specifically pertaining to people with complex class III obesity.

One of the available dietary treatments for complex class III obesity is a Severely Energy-Restricted Diet (SERD). SERD is a collective term used to encompass diets involving severely restricted energy intake relative to energy requirements. SERDs include Very-Low Energy Diets (VLEDs) and Low-Energy Diets (LEDs). VLEDs are defined as weight loss diets that provide between 2100 to 3400 kJ (500 to 800 kcal) per day [1, 17, 18]. LEDs are diets that provide between 4200 and 5000 kJ (1000 and 1200 kcal) per day [19]. VLEDs and LEDs can be achieved with (i) sole use of meal replacement products, (ii) a complete food-based prescription, or (iii) a combination of the two, as a partial meal replacement diet. Commonly, VLEDs are prescribed as a total meal replacement diet and LEDs as a partial meal replacement diet. Whether a VLED or an LED, SERDs are considered as severe dietary energy restriction for people with class III obesity, because they provide an energy intake prescription of $\leq$ 5000 kJ (1200 kcal), which represents an approximately 65% dietary energy restriction relative to estimated total energy expenditure for this group [20, 21].

SERDs involving the use of meal replacement products achieve $\geq$ 10% weight loss when followed for 6 weeks or longer in people with class III obesity [21]. Although this clinically-relevant weight loss is achieved in clinical trials, a recent survey of healthcare professionals found that they only prescribe meal replacement SERDs to a median 7% of all patients seeking weight management [22]. An important barrier to prescription found amongst healthcare professionals was anticipated poor dietary adherence [22], attributed to the highly restrictive nature of the diet, including the extreme reduction in carbohydrate and energy intake [22]. The discrepancy of what is reported in clinical trials and what is reportedly observed by healthcare professionals may be due to clinical trials being vastly different to real-world use, where interactions between healthcare professionals and participants may be less frequent and less comprehensive.

What is not well known in the body of literature is participants' perceptions of SERDs and the potential factors that facilitate or create barriers to adherence. Healthcare professionals' perceptions of participant dietary adherence during a SERD may not align with the actual participant lived experience. Understanding what factors contribute to the differences in these

perceptions is important for shaping clinical care [23, 24]. Such an understanding has allowed services and treatments to be adjusted to better meet the needs of the people seeking treatment [23, 24], including providing insight into the complexities of the participants' conditions and their medical care [25].

Previous qualitative studies that have investigated the experiences of participants with overweight and obesity during total meal replacement VLEDs have shown that they are positively accepted [26, 27]. Facilitators of dietary adherence identified included the diagnosis or presence of serious medical conditions and the desire to improve personal appearance and feelings of well-being [23, 26, 27], the rigid and simplistic nature of the diet [23, 26, 28, 29], supportive group meetings [26, 28], and building personal relationships with dietary counsellors [27]. Less reported are the barriers to adherence during a VLED, which included experiencing challenges with social situations [27], cost of purchasing meal replacement products [26, 27], and lack of social support [28].

In this study we explored the facilitators and barriers to adherence and dietary program attrition in a cohort of people with complex class III obesity who had undertaken a SERD programme at a tertiary weight management service for people with obesity. It is anticipated that our interpretive description will be used to shape the redevelopment of SERD diet programs and assist in treating people with class III obesity more effectively.

## Methods

### Design

This retrospective qualitative study used individual semi-structured interviews. The research design was chosen as an exploratory method to gain an understanding of the facilitators and barriers that contribute to SERD program adherence and attrition. The study was inductive and used for the understanding of factors affecting clinical care rather than theory generation [30]. The study was approved by the Sydney Local Health District Human Research Ethics Committee–Royal Prince Alfred zone (X17-0397 & HREC/17/RPAH/595).

### Participants

The study was conducted at the Metabolism & Obesity Service (MOS), Royal Prince Alfred Hospital, a public university hospital in Sydney, Australia. Patients are referred to MOS by their treating physician. A process of referral triage is undertaken so that patients are invited to attend the most appropriate clinical care stream. The sample of participants studied was chosen from a pre-existing clinic specifically for people with complex class III obesity. In this clinic, individuals are prescribed a SERD and the diet program is delivered in a group setting.

Participants were eligible for participation in this study if they began the SERD program between January 2016 to May 2018, had attended the initial 60-minute SERD group session and at least one visit to the SERD support group or one other individual appointment with a clinician from MOS. The diet program prescription was provided in the initial group session.

The SERD group program was facilitated by a clinician with extensive experience in behavioural health coaching and prescription of SERDs. Participants were instructed to attend the SERD group program fortnightly for education, support, monitoring and to aid adherence. Fortnightly group sessions included 60 minutes of participant-led group discussion, impromptu nutrition and exercise education and behavioural health coaching, led by the group facilitator. The SERD included the prescription of commercially-available VLED meal replacement products supplemented with powdered protein to create a $\geq 65\%$ total daily energy restriction while maintaining an average recommended daily intake of protein of approximately 0.8 g per kilogram of ideal body weight for all participants (mean target based

on 0.84 g per kilogram of ideal body weight for men, and 0.75 g per kilogram of ideal body weight for women [31]).

Participants were provided with a list of suitable commercially-available VLED meal replacement product brands which varied in price, flavour and texture. These entailed meal replacement products from Nestlé Australia Ltd, Tony Ferguson Wellness Program Ltd, Healthy Weight for Life Ltd, Formulite Pty Ltd, Cambridge Weight Plan Ltd, and Optislim Pty Ltd. Participants were also provided with a list of suitable supplemental protein powder brands, namely: Nestlé Australia Ltd Beneprotein; Body Science International Bsc Pty Ltd Whey Protein; Vitaco Health Ltd Aussie Bodies Whey; Freedom Foods Group Trading Pty Ltd Vital Strength Protein Powder; and Vitaco Health Ltd Musashi 100% Whey. The method used to calculate the appropriate protein intake for participants was adapted from Gibson et al., 2016 [32]. In this calculation, 0.25 kilograms was accounted for every 1 kilogram increase in body weight above calculated weight at BMI = 25 kg/m$^2$, using the following formula;

$$\text{Adjusted body weight} = ((\text{height}^2 \text{ x } 25) + (0.25 \text{ x } (\text{current weight} - (\text{height}^2 \text{ x } 25))$$

The estimated individualised protein requirement was then calculated using the following formula;

$$\text{Protein requirement (grams per day)} = \text{Adjusted body weight x } 0.8$$

The calculated protein requirement was used to determine the number of meal replacement products and amount of supplemental protein prescribed per day. Typically, three to four meal replacement products were prescribed per day, with each meal replacement product containing approximately 20 g of protein. At the intial group appointment, participants were given a limited choice of how many meal replacement products they preferred to use in combination with supplemental protein. For example, if a participant's protein requirement was 90 g daily, the prescription could either be 3 meal replacement products (with 60 g of protein) plus 30 g of protein from the supplemental protein source per day, or alternatively, 4 meal replacement products (with 80 g of protein) plus 10 g of protein from the supplemental protein source per day.

If a participant's protein requirements exceeded that which could be achieved using 4 meal replacement products per day, with or without the addition of supplemental protein, or if a participant expressed a strong desire to eat food while on the SERD, food-based protein was prescribed instead in a partial meal replacement format. For example, a selection of protein-rich food sources was prescribed in specific quantities for one or two meals per day (approximately 100 g lean red meat, 200 g chicken, 300 g white fish, 100 g oily fish, 200 g pork, or 3 eggs) and a mixture of low-starch vegetables from a provided list. The prescribed high protein meal approximated to 20 to 30g of protein and an energy intake (EI) of 837 to 1050 kJ (200 to 250 kcal).

Participants were instructed to consume 10 g of fat, a minimum of 2 L of water, and a minimum of 5 standard serves of low-starch vegetables (with one serve equating to 1 cup of salad vegetables or ½ cup of cooked vegetables) daily for a minimum of 3 months. The total estimated daily intake of the SERD prescribed ranged from ~3000 to 4600 kJ (700 to 1100 kcal) for the group of participants sampled.

After program completion or unplanned early cessation, participants were given autonomy and support to either continue with the meal replacement SERD, to transition to a moderately energy-restricted diet with partial use of meal replacement products, replacing 1 to 2 meals per day together with protein-rich food for the third meal, or to transition from the SERD to an

energy-controlled food-based diet. The weight management service supported people long-term (>12months) to help them reach their weight loss goal.

## Recruitment

In August 2018, eligible participants (n = 53) received a letter stating that they may receive a telephone call regarding study recruitment. One researcher (G.M.) invited participants by telephone in September and October 2018, and in January 2020, to participate in an interview. Written informed consent was obtained during in-person interviews. Verbal informed consent was used when written consent was not obtainable, such as when the participant had moved interstate or could not attend in-person appointments due to time, geographical or financial constraints. Verbal consent was obtained at the initiation of the audio-recorded telephonic interviews and documented in the interview transcript. This methodology was approved by the Sydney Local Health District Human Research Ethics Committee–Royal Prince Alfred zone.

From the group of eligible participants, participants were then recruited through purposive sampling [33] by the lead researcher, G.M. This method is used commonly in qualitative research and includes the deliberate selection of individual participants because of the information, knowledge or experiences they possess [33]. After reviewing electronic medical record data on weight history and clinic attendance, three types of participants were contacted; those that (1) had experienced and maintained a weight loss of >10% at the time of recruitment, (2) had not completed the 3-month duration of the SERD group sessions, or (3) had recorded no change in body weight from baseline at the time of recruitment. Interviews were then conducted 1 to 2 weeks after the initial phone call. After preliminary data analysis of the interview was completed, further participants were recruited and interviewed until saturation in the data was confirmed for each of the reported themes. After saturation, a follow-up letter was sent to the remaining participants to inform them that recruitment had ended.

Eighteen participants were contacted and recruited between September and October 2018, and the remaining two were contacted and recruited in January 2020. This delayed recruitment method of the last two participants was used to confirm saturation after the initial data analysis, to ensure the identified themes could be applied to new interview recruits. No incentives were offered for participation.

## Data collection

Initially, participants were asked to participate in the interview in a group setting, however, due to low attendance, this data collection strategy was revised to one-on-one in-depth interviews either by telephone or in-person. The initial group-based interview consisted of two participants and was carried out by two researchers (G.M. and J.F.) at the hospital weight loss service location. The remaining eighteen participants were interviewed individually by telephone or in-person by one researcher (G.M.) at the participants' convenience. G.M. is an experienced dietitian completing her doctoral degree in research with limited experience in qualitative research, however, guidance was provided by co-authors who had some experience in qualitative research (A.S., J.F. and J.S.). G.M. was known to 15 of the 20 (75%) participants, but was neither their primary clinician nor SERD program group facilitator. Semi-structured interview questions were formulated with open-ended questions to elicit detailed descriptions of participant experience in the SERD program. Using prior knowledge from clinical practice and past research [34], the following themes were explored: past dieting experiences, perceived program adherence and factors that aided/hindered adherence, hunger, current dietary patterns, motivation to lose weight, social environment, self-efficacy, weight-related stigma,

socioeconomic limitations, and service delivery. At the end of the interview, participants were invited to share any comments they considered were not adequately covered during the interview. An interview guide containing the interview questions can be found in the S1 File.

The audio-recorded interviews were transcribed into text verbatim and participant transcripts were allocated pseudonyms. Pseudonyms were chosen to reflect the gender of the participant interviewed. Field notes were taken after the interviews and later coded with the interview transcripts. Medical records were reviewed to obtain participant demographics and characteristics including age, residential postcode, whether they received social security payments (age pension, unemployment or disability support), height, weight and comorbid conditions. To determine the degree of complexity of the participants' obesity, the Edmonton Obesity Staging System (EOSS) tool was used [35]. EOSS scores are as follows: 0 = no sign of obesity-related risk factors, 1 = presence of subclinical obesity-related risk factors that does not require medical treatment for comorbidities, 2 = established physical or psychological obesity-related comorbidities requiring medical intervention, 3 = significant obesity-related end-organ damage or significant psychological symptoms or physical disability whereby quality of life is impacted [35].

Socio-economic disadvantage was determined by the presence of social security payments and residential postcode. The residential postcode was used to determine relative socio-economic disadvantage through the use of the Australian Socio-Economic Indexes for Areas (SEIFA) Index of Relative Socio-Economic Advantage and Disadvantage (IRSAD) [36, 37]. The SEIFA IRSAD is limited in its ability to identify individual socio-economic differences within a residential area mainly due to the distribution of public housing in Australia in both affluent and disadvantaged areas [36], thus social welfare payments were used together with SEIFA IRSAD to identify socio-economic disadvantage across participants. IRSAD SEIFA scores are as follows; 1 indicates the most disadvantaged area and 4 the most advantaged area [36]. A score of 2 or 3 is given for an area that is neither particularly advantaged nor disadvantaged.

## Data analysis

Qualitative interview transcripts underwent thematic content analysis by researchers (G.M. and J.F) who independently reviewed two transcripts using manual line-by-line open-coding to generate a list of inductive codes. The coding lists were combined to create a preliminary list of inductive codes to apply to the remaining transcripts, with new codes generated as new themes and concepts emerged. NVivo v.12 software was used to analyse the transcripts and to group participants' reflections and concepts into unified themes using an iterative process via constant comparison. Constant comparison was used to reduce personal bias influencing data analysis. Transcripts were summarized, and memoing was used to link codes with themes and concepts [34]. After the initial analysis of all the transcripts, three researchers (G.M, J.S and J.F.) reviewed the expanded code list, themes and sections of the data, and discussed their contexts to resolve discrepancies. Subgroup analyses were then performed comparing the responses from participants who experienced facilitators and barriers to adherence, and those who had not completed the program or ceased the SERD prematurely. Participants were not invited to comment on findings. The results in italics are verbatim quotations from participants identified by a pseudonym. The amount of weight change exhibited by each participant at the time of interview has also been provided after each quote (e.g., Adam, 26.2% loss in body weight). Brackets () have been used to provide relevant information that was implied but not spoken. Irrelevant information has been removed and marked with an ellipsis (. . .).

## Results

### Participant characteristics

Twenty participants were interviewed (12 females and 8 males), mean BMI at baseline was $63.7 \pm 12.6$ kg/m$^2$. Six in-person interviews were conducted in total, including the those that were performed in a group setting, the remaining were conducted by telephone. The duration of the in-person interviews ranged from 20 to 75 minutes, the duration of the telephone interviews ranged from 20 to 50 minutes. The average body weight reduction experienced by the group was $9.9 \pm 10.7\%$ at the time of interview. Eleven of the 20 participants (55%) experienced and maintained >10% weight loss at the time of interview.

Socio-economic disadvantage was prevalent among study participants, with 45% (9 of 20) receiving social welfare payments, and 25% (5 of 20) residing in a socio-economically disadvantaged area determined by an overall SEIFA IRSAD score of 1. In combination, 65% (13 of 20) of participants were identified as 'socio-economically disadvantaged'. Detailed participant characteristics can be viewed in Table 1.

The following obesity-related co-morbidities were observed from participant medical records: obstructive sleep apnoea (65% of participants), depression (40%), hypertension (35%), osteoarthritis (35%), type 2 diabetes (30%) and hyperlipidaemia (10%). EOSS scores can be viewed in Table 1.

### Thematic content analysis

The analysis identified 10 themes, five as facilitators and five as barriers to adherence, as seen in Fig 1.

### Facilitators of adherence

**SERD program group counselling and psychoeducation sessions.** Participants thought that the educational group format in which the SERD program was delivered was fundamental to adherence because of the knowledge gained through formal and informal information sharing between other participants and the weight loss therapist.

Participants thought the following nutrition education topics were facilitators to dietary adherence: cooking vegetables in palatable ways, the energy and macronutrient composition of commonly eaten food items, food items to consume and avoid, where to obtain meal replacement products at discounted prices and suitable brands. Psychological educational topics that were considered important for facilitating dietary adherence included: overcoming emotional eating, how to respond to food cravings, navigating social eating and implementing relationship boundaries.

This environment of information sharing and understanding the psychological triggers behind their behaviours allowed participants to experiment and experience autonomous self-regulation. It also provided an environment whereby they felt competent in their ability to change behaviours, demonstrating that the SERD program structure fits well within the self-determination theory framework for understanding human motivation [38]. The framework suggests that when the basic psychological needs for autonomy, competence and relatedness are supported, an environment is created whereby the adoption of healthy behaviours, or relinquishment of unhealthy ones, is more likely to occur [39, 40].

*But where I found MOS [Metabolism & Obesity Services, the name of the weight loss clinic] to be useful was in their lectures and education with the program. They tell you what foods you can and cannot eat. I now know that I can pig-out on vegetables, but choose the non-starchy*

**Table 1. Participant characteristics at baseline, post SERD intervention and at the time of interview.**

| | Pseudonym | Sex | Age (years) | Social security payments | IRSAD SEIFA ranking of disadvantage by suburb location (1 to 4) | EOSS | Reason for discontinuing SERD | SERD duration (months) | Length attendance at weight loss centre at time of interview (months) | Time from drop out to interview (months) | BMI at baseline (kg/m²) | Weight before SERD (kg) | Overall weight change from prior to SERD to interview kg (%) | Largest weight change during SERD kg (%) |
|---|---|---|---|---|---|---|---|---|---|---|---|---|---|---|
| Experienced and maintained a weight loss of >10% at the time of interview | Adam | M | 56 | Y | 4 | 2 | N/A | 6.9 | 6.9 | N/A | 75.2 | 260 | -68.0 (-26.2) | -68.0 (-26.2) |
| | Bella | F | 49 | Y | 2 | 2 | N/A | 4.0 | 4 | N/A | 69.1 | 147.2 | -14.8 (-10.1) | -19.2 (-13.0) |
| | Errol | M | 45 | N | 1 | 2 | Time and cost of travel to service | 3.0 | 7.7 | 13 | 79.1 | 265 | -15.2 (-5.7) | -15.2 (-5.7) |
| | Franny | F | 52 | Y | 1 | 3 | N/A | 12.0 | 12 | N/A | 65.1 | 175 | -71.3 (-40.7) | -71.3 (-40.7) |
| | Ialia | F | 60 | Y | 2 | 2 | N/A | 14.0 | 14 | N/A | 78.9 | 187 | -30.1 (-16.1) | -30.1 (-16.1) |
| | Jackson | M | 41 | N | 2 | 2 | N/A | 10.0 | 14 | N/A | 63.2 | 191.2 | -37.7 (-19.7) | -37.7 (-19.7) |
| | Peta | F | 43 | N | 1 | 2 | Started working and could not attend daytime clinic times | 6.0 | 24 | 20 | 90.2 | 222.4 | -24.2 (-10.9) | -24.2 (-10.9) |
| | Quinn | F | 60 | N | 3 | 2 | N/A | 1.0 | 14.1 | N/A | 57.3 | 143.1 | -18.9 (-13.2) | -7.8 (-5.5) |
| | Rachel | F | 52 | N | 3 | 2 | N/A | 7.0 | 21 | N/A | 52.5 | 139.4 | -16.0 (-11.5) | -16.0 (-11.5) |
| | Steve | M | 26 | N | 4 | 0 | N/A | 4.4 | 16.2 | N/A | 62.4 | 261.7 | -41.3 (-15.8) | -41.3 (-15.8) |
| | Tom | M | 39 | N | 4 | 2 | N/A | 3.3 | 20.2 | N/A | 63.4 | 244.4 | -36.6 (-15.0) | -36.6 (-15.0) |
| Did not record 10% weight loss at time of interview | Carl | M | 44 | Y | 4 | 2 | Cost of travel to service | 6.0 | 17.2 | 5 | 73.5 | 215 | 1.0 (0.5) | -15.0 (-7.0) |
| | Diana | F | 61 | N | 1 | 2 | N/A | 5.0 | 23.8 | N/A | 55.8 | 155.7 | -6.9 (-4.4) | -6.9 (-4.4) |
| | Georgie | F | 68 | Y | 4 | 0 | Didn't like the taste of meal replacement shakes | 1.0 | 1.6 | 25 | 76.7 | 198.8 | -1.0 (-0.5) | -1.0 (-0.5) |
| | Harriet | F | 53 | N | 2 | 2 | N/A | 6.0 | 6.2 | N/A | 48.7 | 139.1 | -7.5 (-5.4) | -9.1 (-6.5) |
| | Kelly | F | 70 | Y | 4 | 2 | N/A | 12.0 | 25 | N/A | 49.5 | 108.4 | 3.1 (2.9) | -10.4 (-9.6) |
| | Louise | F | 57 | Y | 2 | 2 | Experienced self-reported depression & thought she was ineligible for bariatric surgery | 4.0 | 19.9 | 10 | 67.7 | 172.1 | -7.9 (-4.6) | -11.2 (-6.5) |
| | Mark | M | 54 | N | 1 | 2 | Moved interstate for work | 2.5 | 3.9 | 14 | 43.4 | 172.9 | 0.5 (0.3) | -12.9 (-7.5) |
| | Naomi | F | 33 | Y | 3 | 2 | N/A | 6.5 | 11 | N/A | 50.2 | 126.9 | -0.6 (-0.5) | -17.5 (-13.8) |
| | Oliver | M | 60 | N | 3 | 2 | Didn't like the taste of meal replacement shakes Thought the program was not suitable | 0.3 | 23.1 | 25 | 51.1 | 144.6 | -1.1 (-0.8) | -1.1 (-0.8) |
| | **Mean** | | 51.2 | N/A | N/A | N/A | | 5.7 | 14.3 | 16.0 | 63.7 | 183.5 | -19.7 (-9.9) | -22.6 (-11.8) |
| | **SD** | | 11 | N/A | N/A | N/A | | 3.8 | 7.5 | 7.5 | 12.6 | 47.8 | 21.9 (10.7) | 19.7 (9.3) |

**Abbreviations:** M, male; F, female; SEIFA, Socio-Economic Indexes for Areas; EOSS, Edmonton obesity staging system; N, no; Y, yes; BMI, body mass index, SD, standard deviation; N/A, not applicable.

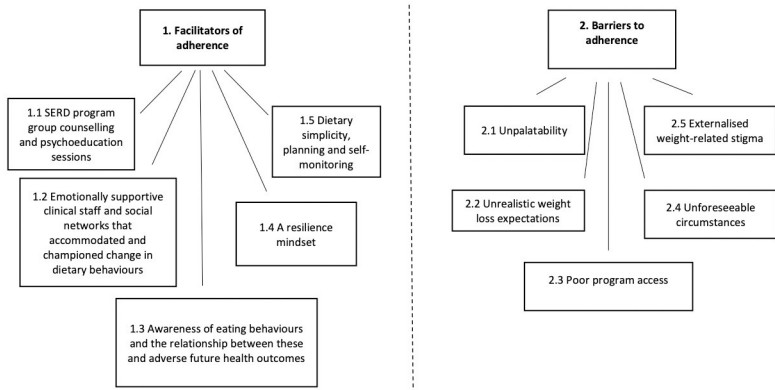

**Fig 1. Facilitators and barriers to adherence.**

*vegetables such as broccoli, cabbage and eggplant. For me, it's about knowing what I can eat, which I didn't know before. (Adam, 26.2% loss in body weight)*

*Having nutrition knowledge also helped. I would look at ingredients and nutrition panels on food when I was shopping. I did a lot of online searching for recipes, using the list of foods that I could and couldn't have. Then I would go looking for recipes that matched those foods and things I could do with them to make it more interesting. (Harriet, 5.4% loss in body weight)*

*My mum and dad being Italian, they would serve me pastas and bread etc, first course, second course, third course [when visiting]. I said to them look, I'm on this diet don't put anything in front of me; if you are going to put anything in front of me put the salad. And my mum knows. I just have my shake with my salad and water. (Bella, 10.1% loss in body weight)*

**Emotionally supportive clinical staff and social networks that accommodated and championed change in dietary behaviours.** Participants described an initial hesitance to attending the weight loss clinic because of previous stigmatising experiences (both medical and social); however, after attending the weight loss service, many considered they were in the right place to receive help. This was important for program adherence and behaviour change as participants felt they could implicitly trust the weight loss therapists to provide the right advice that would be delivered without judgement. The purpose-built physical clinic space including large open spaces in the waiting area, corridors and clinic rooms, and the use of specialised bariatric size chairs and scales without size or weight limit contributed to the feeling of being in a non-judgemental environment. Participants felt a sense of belonging when attending the weight loss service where they were exposed to other larger-bodied people. They felt they could relate to other participants in the waiting room and during SERD group sessions. This allowed participants to feel welcomed and at ease, as they could see peers who were similar to them.

*I felt welcomed when I saw people who were in the same situation to me. I knew there was no judgement. I feel great going down there all the time now. It's like a boost in my confidence each time I go into the clinic. (Jackson, 19.7% loss in body weight)*

*I went to a clinic for bariatric surgery many years ago. I am guessing people who search bariatric surgery would be obese, well most of them anyway. But when I got there I couldn't even sit in my chair. I got there and thought, what kind of place is this? . . .The clinic obviously didn't think about the size of the person they see. What I liked about the obesity clinic [Metabolism*

*and Obesity Service] is you have big chairs for big people. That makes sense. (Errol, 5.7% loss in body weight)*

*I remember my first time coming here, it was nice. It was nice to know, not that I want other people to have trouble with their weight, but it was nice to know that other people had troubles with their weight and just all coming together we had a shared experience. You don't feel alone with it. It's nice to come and be able to talk about things, find out how this person struggled, and that person struggled and how they got through it, it just helps. (Steve, 15.8% loss in body weight)*

Participants discussed the importance of having support from friends and family to aid adherence when eating outside of the home and at social functions. Others felt adherence was facilitated by family members who kept the home food environment free from discretionary foods such as chocolate, biscuits, crisps and soft drink. When eating outside of the home, support was felt when participants were able to modify social eating situations, such as having a degree of control over what food items were served or purchased. Participants felt more supported when other people altered their eating patterns to healthier meal options as this reduced their temptation to eat energy-rich food. For participants with low-income backgrounds, financial support from family members was often integral to adherence. This removed accessibility barriers such as the cost of purchasing SERD products and transport to the weight loss service that would have otherwise made participation in the SERD program unaffordable. Verbal encouragement from weight loss staff, friends or family was conducive for adherence and provided reassurance that they were making good progress, which bolstered their motivation to continue.

*My friends, they are very supportive, extremely supportive. They are happy to go where I suggest. I ask them can I choose where we eat? And they are happy to go with it. I just tell them I can have something grilled. (Quinn, 13.2% loss in body weight)*

*My family understood whenever I was with them, say I was going to my daughters for lunch, she would make me nice steamed vegetables because she knew I could have that with the meal replacement. (Ialia, 16.1% loss in body weight)*

**Awareness of eating behaviours and the relationship between these and adverse future health outcomes.** Before joining the SERD program 85% of the participants had previously tried to lose weight using other dietary weight loss methods with varying levels of success. Some of the reasons for abandoning previous dieting attempts included the slow weight loss experienced and falling into old patterns of behaviour. In contrast, during the SERD program, the speed of initial weight loss experienced during the first two weeks of the diet was integral to facilitating motivation to adhere to the program. Minimal initial weight loss contributed to poor adherence and/or attrition because participants felt that their efforts were futile.

Alleviation of medical problems was also motivating and reaffirmed why participants were engaging in the SERD program. Biofeedback, such as feeling ill after eating discretionary food or lack of weight loss observed during the diet, was used to correct patterns of behaviour to aid adherence. For example, participants acted on advice to i) change how they shopped for food to avoid purchasing tempting items at the supermarket, ii) cook vegetables in a large quantity in advance to avoid ordering takeaway and iii) carry meal replacements bars with them when outside of the home to avoid purchasing discretionary food when hungry.

*With the program I thought, I tried dieting and everything in the past and I'm just going to fail and when I started seeing the results using the shakes, and then got the confidence from*

*you guys to start exercising. . . it's changed everything. I think the turning point was when I stuck by it for a week and then seeing the results were ridiculous, and I was like wow*! *I kept taking them, got used to them and now I enjoy them. . . .The weight loss was huge, I lost 7kg in the first week and that was without exercise. That was just with the shakes. It was a massive factor and the main reason I stuck with them. (Jackson, 19.7% loss in body weight)*

*My depression is linked to my weight, so when my weight deteriorates, so does my depression. When I started to gain weight, that's when my depression came along too. That's when I started to become demotivated to do anything really. That's where I found the drop-in group sessions extremely helpful. I did my weigh-ins whenever I went there, every 2 weeks, so I could see the weight change and also so I could focus on the strategies to help me eat right and talk about the things that made me eat the wrong things. Why my diet was so poor in the first place. (Tom, 15.0% loss in body weight)*

Personal awareness allowed participants to make the connection between their state of obesity and how it would affect their lives in the future. Participants then harnessed motivation by focusing on personally important goals, which provided strong reasons for behaviour change. This was facilitated by providing participants with knowledge through the group sessions in which competence to manipulate food types was gained and non-judgemental support was provided from the group facilitator and group members. Some of the personally important goals included being able to take care of disabled or young family members, alleviation of some co-morbid disease symptoms, a degree of pain relief and experiencing independence such as walking or being able to travel overseas unassisted. These personally important goals were recalled at critical points when adherence would have usually declined. Critical points included when challenged by feelings of hunger, weight-related stigmatising situations that led to low mood or during social eating occasions.

*I want to be able to breathe. That was my main thing, in the beginning, is I couldn't breathe and that really scared me. Because I knew I stopped smoking 20 years earlier, then I thought it was my heart and I went to doctors and it wasn't my heart. Then I went to a lung specialist, who told me it was my weight crushing my lungs. . .and I thought is this what you want for the rest of your life? Be one of those disabled people in a power shopper [electric powered disability scooter]. (Franny, 40.7% loss in body weight)*

My wife and my daughter were my main motivators and then getting rid of the reflux. My reflux was so bad I couldn't even have a glass of water without regurgitating it back up. . . You have to do it for yourself and you have to do it for your daughter. My daughter is 3, I'm scared that I wasn't going to be around if I stayed on the track I was on. (Jackson, 19.7% loss in body weight)

*On a bad day [a day on which the diet wasn't followed], I thought I'd had enough. . . it's because I felt like I was getting made to do this. I had that negative thoughts- why should I be made to do this? After realising that eating like the old me made me feel off, it was not that everyone was making me do it, it was now my choice. I want to do it. (Ialia, 16.1% loss in body weight)*

**A resilient mindset.** Participants who experienced positive dietary adherence demonstrated mental resilience around setbacks. Setbacks included experiencing challenging social eating occasions whereby extra food was consumed or experiencing negative comments from others prompting emotional eating. Mental resilience included positive self-talk, reminding

themselves of their achievements and their long-term personally important goals. Weight loss was re-framed as a long-term incremental process in which even the smallest amount of weight loss and weight stabilisation were celebrated. Celebrating small achievements such as these helped to circumvent feelings of disappointment and failure, which was acknowledged to be the cause of complete dietary abandonment in past dieting experiences.

> *You feel like I've started at such a high weight, this small amount of weight loss is just a drop in the ocean. . . I stopped worrying about goals, like a certain amount of weight in a certain amount of time. My goal is to just lose weight, that's it. When I make concrete goals there's too much pressure on me and then I fail. So now there are no expectations, no timelines. As long as I lose I lose and that's it. (Errol, 5.7% loss in body weight)*

> *The diet was working for me because I was losing weight, albeit slowly. It wasn't in the way that I had read about, you know how you're supposed to lose heaps of weight when you go into ketosis. . .but I thought, instead of being disappointed because the weight is coming off slowly, think of it positively. The weight is still coming off and it's probably better to come of slowly. (Quinn, 13.2% loss in body weight)*

> *I'm still getting negative comments from my family like [about following the diet] 'oh stuff that', 'what do you mean you can't eat that? Don't be ridiculous', 'oh surely you can eat this?'. But when you finally lose weight its 'gee you're looking good'. Yes, because I'm sticking to it and not listening to what you lot are saying. (Franny, 40.7% loss in body weight)*

**Dietary simplicity, planning and self-monitoring.**   The highly restrictive nature of the SERD meant there were fewer foods options from which to choose, fewer opportunities to eat and reduced food availability in the home. The use of a formula product in a portion controlled sachet seemed to be an important component to the perception of dietary simplicity. The use of meal replacement products reduced the frequency of supermarket visits, limiting exposure to problematic tempting food items. Thus, participants felt the SERD was easier to follow and less cognitively demanding than dieting with unrestricted food choice.

> *I think it was the continuity of the program, what I had to cook and what I had to take to work, and simplicity of the diet. At that stage, you aren't going to the shops and walking past the biscuit aisle and saying "oh, I just want that" and put that in and that in [the trolley]. You go to Chemist Warehouse and you've already bought it [the meal replacement products at the chemist], so those temptations aren't there. (Rachel, 11.5% loss in body weight)*

> *I found the diet very practical because it's in the sachets. If it was a diet where I had to eat this, then eat that and having to manage that, it would be too hard. (Adam, 26.2% loss in body weight)*

> *. . . with the shakes it's much easier to maintain, it's a simpler diet to maintain. It's been 2.5 months for me now on the shakes and once you're in a routine it's much easier to stick to it. (Tom, 15.0% loss in body weight)*

Although the SERD was viewed as simple and easy to follow, participants needed to plan for every step of the food preparation process to facilitate the implementation of the diet. For example, participants would search for vegetable-based recipes to cook for the subsequent week, create shopping lists to avoid buying additional food items, make conscious decisions to avoid certain aisles of the supermarket ahead of time, avoid the supermarket entirely by shopping online or at smaller food stores, purposely carry meal replacement bars in their car or

bag, or review restaurant menus ahead of time to find suitable options. These various methods of planning facilitated participants' adherence to the SERD long-term.

> *Just be prepared, having meals prepared and planning your day and week. . . Make sure you have the right foods at home. You see that was a big problem for me. Times when you are in a rush, you can't go to takeaway stores because that's all the bad food. . .so taking stuff with you helps. (Peta, 10.9% loss in body weight)*

> *I found if I skipped meals my hunger would come like double. I just want to destroy everything in front of me, it's a self-taught thing you have to do to work out how to get through it. You have to work out the timing and planning of your meals around exercise to work out when to have it [the meal replacements]. . .The routine is the thing you need to get for a while. (Errol, 5.7% loss in body weight)*

> *I think you get into a routine or structure. Once you get into the structure or routine, you say to yourself–well I liked that, I'm going to cook that one again [the vegetable recipe]. (Rachel, 11.5% loss in body weight)*

Monitoring weight loss assisted with motivation to adhere to the SERD program. Regular weigh-ins were a vital component of the SERD group sessions as it was used by the participants to validate their efforts. If the weigh-in did not result in weight loss, participants reported feeling disappointed, but this situation was then used as an opportunity to discuss behavioural barriers associated with lapses in adherence with the group facilitator and other members of the SERD group. The discussions were used to resolve how to overcome barriers participants had experienced in the past few weeks. Feelings associated with disappointment seemed to be resolved following the group discussion. Some participants did acknowledge temporary lapses in adherence when they weighed in at home without support. In these instances the prospect of turning up to the next weigh-in and group session was used as a motivation to resume the diet.

Monitoring exercise also assisted with SERD program adherence. Participants reported experiencing an increased rate of weight loss when exercising. Thus, exercise was viewed as a vital component of the SERD program to improve feelings of well-being, reduce disability and maximise the weight loss experienced.

> *I hopped on the scales every couple of days and kept track of it. It was a very good motivation. (Bella, 10.1% loss in body weight)*

> *With the exercise as well, when I started this I could only do 1000 steps for a walk. Now I'm doing a minimum of 10,000 steps just in my morning walk. This is massive. When I first started, I did one school block. I was so tired, I couldn't breathe. Now I'm doing 7 km in one morning walk. I feel great! Now when I don't get to walk I get frustrated. (Jackson, 19.7% loss in body weight)*

> *The group sessions are helpful and I coincided my weigh-ins whenever I went there, so every 2 weeks. Number 1, because the scales I need, need to be at an obesity clinic because I'm too heavy. It was a good way to break it down in those 2 week increments, not to focus on the actual weight coming down but focus on the strategies to help me eat the right things. (Steve, 15.8% loss in body weight)*

## Barriers to adherence and SERD attrition

**Unpalatability.**    Participants who found the taste and texture of the meal replacement products unpalatable found adherence challenging. Participants did not engage with product

experimentation to find palatable brands to purchase because they did not have the financial means to do so.

> *When I was doing the program the biggest problem I had was the flavour and texture. The texture had a lot to do with it especially at week 7 or so, I really started craving crunchy textures. (Harriet, 5.4% loss in body weight)*

> *It was difficult to afford to buy the shakes. . .I felt frustrated because I couldn't do them [the meal replacement shakes]. But I did like the program. (Peta, 10.9% loss in body weight)*

**Unrealistic weight loss expectations.** Unrealistic weight loss expectation was another barrier to adherence. Participants experienced a mismatch between what they perceived as expected weight loss and what they were able to achieve. Participants then perceived their short-term efforts in adhering to the SERD as insurmountable to the weight loss goal they had set for themselves long-term. This then led to partial or complete dietary abandonment, which exacerbated the frustration felt by a lack of extreme weight loss.

> *I never really felt satisfied on the shakes. But then I am not a chronic eater. I can't remember what the scales where doing at the time. . . I think it came down only 1kg. I did one shake a day for 2 to 3 days and just gave up. This is just going to take too long. (Oliver, 0.8% loss in body weight)*

> *I'm pretty erratic with it [the SERD] because that tends to be my personality. . . in the first 12months I started at 215kg and I got down to 200kg and I went back up to 211kg. I kind of feel like it wasn't a complete waste of time. But it's hard you know, some days I just give up. I still try, and it's important that I keep trying. I did want to get to 90kilos by my brother's wedding [in 6months time]. (Carl, 0.5% gain in body weight)*

**Poor program access.** Poor program accessibility encompassed three areas: (i) the distance from which the participant resided from the weight loss service, (ii) the distance of the weight loss service clinic building from the public car park and public transport, and (iii) financial status.

Poor program accessibility was experienced by participants who lived more than one-hour travelling distance to the weight loss service. The distance participants needed to travel to attend the program affected group attendance because of the cost and time associated with public transport or parking one's car. The cost and time needed for travel to the weight loss service were amplified by the physical limitations of the participant. Older, larger-bodied and participants with a disability found the distance needed to mobilise from the car park to the weight loss service a barrier to program attendance. Irregular attendance contributed to participants feeling they lacked accountability and support needed to effectively follow the SERD program.

> *The thing I didn't like was the parking. We are big people, we can't park 2 kilometres down the road and walk 2 kilometres to the clinic [approximate distance to the clinic is 300 meters from the public car park and to on street parking]. These little things deter you and the obstacles that stop you from coming. And that's what got me eventually, it was just too far. . .I have to come there at least half an hour early and drive around like a crazy person and then you park anywhere and you come back and get a fine, or pay for parking and it's not cheap. (Errol, 5.7% loss in body weight)*

*Coming in that far was a real chore and expense as well. But I kept it up as much as I could. I managed to get some extra government support financially, which kept me going for the first couple of months. . . It takes 1 to 2hours to drive in, and financially, petrol and parking its tough. I can't park in the parking station where you get a cheaper rate, because I can't walk that far. (Louise, 4.6% loss in body weight)*

Periods of financial instability contributed to transient and complete periods of dietary abandonment. This was because participants did not have the financial means to pay for transport to the weight loss service and were also limited in their ability to purchase the VLED meal replacement products. Financial instability limited the range and variety of meal replacement products that could be purchased. One participant described a period of homelessness during the SERD program, which resulted in discontinuation of the SERD and non-attendance to the weight loss service.

*Recently we had a personal experience where we became homeless, that made me fall off track and I wasn't able to concentrate my mind was elsewhere. I forgot all about the program and I had nowhere to cook. We were living in a motel and just living off what we could. It was very hard money-wise. (Naomi, 0.5% loss in body weight)*

*Sometimes it's hard, like today, it's hard because I had asthma. Today it cost me $100 round trip in the Uber that my parents pay for me, so it's a commitment. Both in the effort and money to get here. (Carl, 0.5% gain in body weight)*

*. . .financially it was a bit tight. It was a bit difficult to buy vegetables with the shakes to make up the meals, and buy the protein on top of that. (Kelly, 2.9% loss in body weight)*

**Unforeseeable circumstances.** Unforeseeable circumstances accounted for the majority of reasons for attrition by participants, and the main factor leading to long periods of poor dietary adherence. Apart from the aforementioned period of homelessness, other unforeseeable circumstances included a sudden exacerbation of conditions such as depression, a cancer diagnosis of a loved one, changes in living arrangements, injury and accident. These events often led to episodes of self-reported depression and anxiety, impaired physical health and disability. For participants who described themselves as emotional eaters, this resulted in temporary dietary lapses.

*My legs, the skin got a scratch on it and it wouldn't stop bleeding, like water. I was losing all the skin on my legs. I had an ulcer. I had to get some meat or iron back into me to help the healing. That seemed to be the only thing that worked. So, I went down to 2 shakes and a meat in between. Then my daughter and I had a car accident in January and I tore my shoulder. The pain killers and feeling useless, just put me back on to food. The shakes weren't working with the tablets, I started getting dizzy spells. (Mark, 0.3% gain in body weight)*

**Externalised weight-related stigma.** Dietary lapses and emotional eating due to weight-related stigma were openly discussed by participants during the interview. Externalised weight-related stigma were most notable when they felt they were the most vulnerable, such as during exercise, episodes of poor mental health or when outside of the home shopping. Feeling stigmatised by outsiders because of their body weight and size led to periods of low mood and emotional eating. When this occurred during their SERD program it contributed to poor adherence and/or dietary attrition. All participants, regardless of weight loss outcomes, perceived themselves to be mostly motivated and adherent to the diet, suggesting there was a

degree of poor self-awareness in some participants. The action of turning up to appointments and participating in the group sessions were perceived by the participant as evidence of their motivation, even when weight loss was not achieved.

> *I think it [feeling stigmatised] did affect my ability to try weight loss because I thought; I don't have to justify anything to anyone else. Who are you to judge me? And that's kind of the downfall. You walk around thinking, it doesn't even matter if I am trying, people are going to look down at me anyway. What's the point in trying? (Errol, 5.7% loss in body weight)*

> *Stigma did affect me by feeling really upset. I didn't want to go out. People look at you like you're an alien at the shops, so I socially isolated. That made me start emotional eating again and that made me put on more weight. Then my doctor put me on to this program. Because of all the negativity and the way people looked at me, the first couple of times I went down to the clinic I was embarrassed to be there. My family had to push me to go because I didn't want to be there. I was too embarrassed. (Naomi, 0.5% loss in body weight)*

## Discussion

Our findings add to the evidence base by demonstrating that specifically during a SERD meal replacement program, poor program accessibility due to socioeconomic disadvantage, the palatability of meal replacement products and unforeseeable circumstances are significant barriers to adherence and can contribute to program attrition.

Socio-economic disadvantage was prevalent (65%) among participants with complex class III obesity, which is representative of the patient population attending the weight loss service. In Westernised countries such as Australia, socio-economic disadvantage has an inverse relationship with body weight [41, 42]. The most disadvantaged groups have a greater risk of obesity [43, 44], and poor physical and mental health [45–47]. Class III obesity affects those who are more socio-economically disadvantaged to a greater extent than other strata [48, 49]. It was evident by our findings that socio-economic disadvantage was an underlying factor that interferes with SERD dietary adherence and promotes program attrition in this population in many ways. It directly affected program accessibility through inhibiting the ability to experiment with meal replacement product brands when palatability issues occurred and to attend the weight loss centre due to the cost of transport. Socio-economic disadvantage also contributed to many unforeseeable circumstances such as homelessness or changes in living arrangements. Indirectly, it affected those with pre-existing health conditions when they experienced a deterioration in their health, most notably self-reported depression or low mood.

Product palatability, socio-economic disadvantage and unforeseeable circumstances may appear to be non-modifiable barriers; however, the SERD program can be structured to address these issues. Information such as personal demographic factors (residential address and government social support received) can be gathered during participant screening before the commencement of a SERD diet and can be used to determine diet program suitability and to identify those who may require additional support. For example, this may include providing participants with different product samples of brands and product lines to taste before commencement of the diet and asking the participants if there are any transport or logistical barriers to clinic attendance. Additional support can then be provided at low cost by the adoption of telehealth services to reduce barriers related to program accessibility [50] with the partial or complete substitution of in-person appointments to telehealth modes such as telephone calls [51], videoconferencing [52], website use [53], email [54] and mobile phone applications [55]. The combination of in-person and telehealth appointments appears to be a viable and effective

form of weight loss service delivery [54–56]. Healthcare professionals are also the best placed to provide meal replacement product knowledge and recommend alternative product brands at varying price points, flavours and ingredients to overcome issues regarding palatability and dietary adoption barriers.

During this investigation when participants were identified to have palatability and dietary adoption barriers, the option to consume food-based protein was provided. However, participants did not appear to regard the addition of food-base protein as important to dietary adherence. During the interview process, participants were specifically asked about their perception of the effect of supplemental protein and the mode in which it was consumed e.g., powder or via food, on dietary adherence. They were also asked about the key factors they perceived to aid adherence. The inclusion of protein in any form was not identified by any participant in this study as a factor that influenced dietary adherence.

It is challenging to prepare participants for unforeseeable circumstances that may occur during their participation in a SERD program. It is also difficult to predict how and why these events affect some people in certain ways and not others. In these circumstances, efforts could be focused on providing a non-judgemental clinical environment whereby participants are encouraged to return with ease, after such events, to recommence their weight loss journey. This may improve adherence by retaining participants long-term thus potentially facilitating weight loss at subsequent attempts of the SERD program.

The findings of this study also extend the evidence base. We have demonstrated that the barriers found to affect dietary adherence during a SERD among people with class III obesity are also known barriers that affect people with overweight and obesity during most weight loss diet programs. This includes self-imposed unrealistic weight-loss expectations [57], poor program accessibility related to socioeconomic disadvantage (such as the inability to purchase meal replacement products and cost of transport to the weight loss service) [58, 59], weight-related stigma [60, 61] and depression [62]. The cost of purchasing meal replacement products has previously been identified as a factor leading to poor dietary acceptability and poor adherence in two previous qualitative studies [26, 27], and our investigation extends this finding by demonstrating its presence among patients with complex class III obesity. Thus, dietary adherence is affected by many factors, which are unrelated to the type of dietary intervention.

This research confirms previous findings from qualitative research studies that investigated the experiences and the acceptability of total meal replacement VLEDs for participants with overweight and obesity [23, 26–29]. The common facilitating factors to positive adherence are: SERD group counselling and psychoeducation sessions (1.1), emotionally supportive clinical staff and social networks that accommodate and champion change in dietary behaviours (1.2) and dietary simplicity, planning and self-monitoring (1.5).

This exploratory research study has also revealed that there are two factors that may facilitate adherence during a meal replacement SERD that, to our knowledge, have not appeared in the literature before, and these are (1.3) awareness of eating behaviours and the relationship between this and progression of disease and (1.4) a resilient mindset.

Positive dietary adherence requires the adoption of new health behaviours or the relinquishment of unhealthy ones and this is more likely when our most basic psychological needs for autonomy, competence and relatedness are supported [39, 40], which is the premise behind self-determination theory. Autonomous self-regulation and perceived competence were described by participants in the form of the importance placed on the group education, information sharing and the acknowledgement of encouragement received from non-judgmental staff and extended social networks. These facilitators of adherence can be easily fostered in most clinical settings by healthcare professionals experienced in the use of techniques known to promote patient autonomy such as motivational interviewing (MI). MI is a technique used

to guide conversations that focus on participant autonomy and personally important reasons to change behaviour [63] and is effective and feasible at enhancing self-efficacy and dietary adherence in people with obesity [64].

Psychological strategies (including psychoeducation) used for weight management [65, 66] can be incorporated to address barriers to dietary adherence and program attrition including unrealistic weight-loss expectations, externalised weight-related stigma, low mood, and fostering a resilient mindset. Thus, the use of pre-emptive psychoeducation, introduced as routine education provided for all participants, may be one way to reduce participant attrition and discourage weight regain. The list of identified facilitators to adherence could be used to bolster pre-existing, or for new SERD group programs, by assisting in improving adherence during such diets among people with complex class III obesity.

The current study aimed to inform clinical practice, however it is not without limitations. Our study did not determine whether the adherence factors identified were related to the dietary program intervention itself or the pre-existing personality traits of the individual interviewed. There is potential for recruitment bias as those who perceive themselves as less successful or non-adherent may be reluctant to take part in interviews. An attempt was made to capture this group by selectively recruiting those with no weight change and no return appointment until saturation occurred, however it is possible that not all views or experiences may have been captured.

There may be elements of social desirability bias [67], whereby participants provide responses to questions they believe are more socially acceptable, particularly when they feel their responses may be judged or questioned. The participants were aware that the interviews were conducted by a healthcare professional with a speciality in dietetics and may have been conscious of providing 'incorrect' answers with respect to the dietary intervention. Attempts to mitigate this by stating the interviews were confidential and that all forms of feedback were valued to help shape the direction of the clinical service and SERD program, but the degree to which these measures reduced any bias is unknown.

Internalized weight stigma was not a factor that naturally arose during the interview process, it was not asked nor measured via the use of a survey tool. It is possible that internalized weight stigma may be a factor influencing in poor dietary adherence in this cohort. Indeed, a recent systematic review of 10 studies has shown that internalized weight stigma may effect eating behaviours related to weight management [68], however this was not captured in this investigation.

One strength of our study is that participants were recruited from hospital historical records over two years. This ensured we interviewed a range of participants with different experienced who followed the SERD program in the short and long-term and had various outcomes on the diet. We envisage that the research findings will be transferable to a wide range of people with complex class III obesity undergoing SERD weight loss attempts in specialized outpatient weight loss services. It also provides a unique perspective on a population that has been poorly researched and extremely needy of effective care. Key themes generated and identified in this research project will be used to strengthen specialised weight management services and their programs by providing adaptations to educational content and healthcare professional service delivery.

## Conclusion

This study provides a unique insight into the thoughts, experiences and behaviours of people with class III obesity who have participated in a SERD program. The findings highlight opportunities where SERD programs can be optimised to facilitate dietary adherence, thus

improving weight loss outcomes with such programs. This includes the implementation of a SERD program that encourages autonomy, competence and relatedness through education, a supportive non-judgemental environment including the use of external social networks for support. During the initial medical screening, healthcare professionals facilitating SERD programs could screen or identify potential barriers to adherence, which may lead to dietary program attrition. This includes the presence of product palatability issues, unrealistic expectations, low mood, socio-economic disadvantage and externalised weight-related stigma. To address some of these issues psychological techniques that are known to encourage motivation and engagement can be used. Access barriers can be improved by the adoption of tele-health services to deliver dietary program content and to maintain participant engagement.

## Supporting information

**S1 File. Semi-structured focus group and telephone interview questions.**
(PDF)

**S2 File. Participant transcripts.**
(PDF)

## Acknowledgments

We thank psychologist Alexandra Lonergan for assistance with this study.

## Author Contributions

**Conceptualization:** Gabrielle Maston.

**Data curation:** Gabrielle Maston.

**Formal analysis:** Gabrielle Maston, Janet Franklin.

**Investigation:** Gabrielle Maston.

**Project administration:** Gabrielle Maston.

**Software:** Gabrielle Maston.

**Supervision:** Janet Franklin, Samantha Hocking, Jessica Swinbourne, Amanda Sainsbury, Tania Markovic.

**Writing – original draft:** Gabrielle Maston.

**Writing – review & editing:** Janet Franklin, Samantha Hocking, Jessica Swinbourne, Alice Gibson, Elisa Manson, Amanda Sainsbury, Tania Markovic.

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
