## [Decision Letter · Decision Letter 0]

1 Mar 2021

PONE-D-20-39574

Dietary adherence and program attrition during a severely energy-restricted diet among people with class III obesity: a qualitative exploration

PLOS ONE

Dear Dr. Maston,

Thank you for submitting your manuscript to PLOS ONE. After careful consideration, we feel that it has merit but does not fully meet PLOS ONE’s publication criteria as it currently stands. Therefore, we invite you to submit a revised version of the manuscript that addresses the points raised during the review process.

Please address the reviewer comments.

We look forward to receiving your revised manuscript.

Kind regards,

Shahrad Taheri

Academic Editor

PLOS ONE

Journal Requirements:

2. Please provide additional details regarding participant consent. In the ethics statement in the Methods and online submission information, please ensure that you have specified what type you obtained (for instance, written or verbal; if verbal, explain why this was necessary, how it was documented and witnessed, and whether this method was approved by the ethics committee).

3. Please include a copy of the interview guide used in the study, in both the original language and English, as Supporting Information, or include a citation if it has been published previously.

"AS is the author of The Don’t Go Hungry Diet (Bantam, Australia and New Zealand, 2007) and Don’t Go Hungry For Life (Bantam, Australia and New Zealand, 2011). She has also received payment from Eli Lilly, the Pharmacy Guild of Australia, Novo Nordisk, the Dietitians Association of Australia, Shoalhaven Family Medical Centres, the Pharmaceutical Society of Australia, and Metagenics, for presentation at conferences, and served on the Nestlé Health Science Optifast® VLCD™ Advisory Board from 2016-2018. TM has received payments from Novo Nordisk for seminar presentations at conferences. She has served on the Egg Nutrition Council for Australian Eggs since 2004 and Nestle Health Science VLCD Advisory Board since 2010. JF has received payments from Novo Nordisk, Illy, Boehringer Ingelheim and Nestle Health Science for seminar presentations. She served on the Nestle Health Science VLCD Advisory Board between 2009-2013. AAG has received payment from the Pharmacy Guild of Australia and from Nestlé Health Science for oral presentations at conferences. SH has received honoraria from Novo Nordisk, Inova, Sanofi, Eli Lilly, Boehringer Ingelheim and Astra Zeneca for seminar presentations. She has received research funding from Novo Nordisk. GM is the author of The Perfect Juice (New Holland, 2016) and received payment from Novo Nordisk for seminar presentations."

Reviewers' comments:

Reviewer's Responses to Questions

**Comments to the Author**

1. Is the manuscript technically sound, and do the data support the conclusions?

Reviewer #1: Yes

Reviewer #2: Yes

2. Has the statistical analysis been performed appropriately and rigorously? 

Reviewer #1: Yes

Reviewer #2: N/A

3. Have the authors made all data underlying the findings in their manuscript fully available?

Reviewer #1: Yes

Reviewer #2: No

4. Is the manuscript presented in an intelligible fashion and written in standard English?

Reviewer #1: Yes

Reviewer #2: Yes

5. Review Comments to the Author

Reviewer #1: Thank you very much for your recent submission titled “Dietary adherence and program attrition during a severely energy-restricted diet among people with class III obesity: a qualitative exploration”. This explored a very topical area of adherence to diets that severely restrict a person’s energy intake. Adherence to any dietary intervention is key to success therefore understanding key drivers is an important aspect that can help clinician better support their patients.

There are a couple of major points that I think would be good to address. The term “severely energy-restricted diets” is not a typical term that is used in the literature. Having read the manuscript it is clear that that are very clearly discussing a Very Low Energy Diets (VLED) By clearly defining this, the reader can see if the paper is relevant to them, therefore please can you consider changing the use of the term SERD to VLED or Formula Total Diet Replacement depending on which term you prefer. This will prevent the reader having to decipher the methodology before knowing if this is relevant to their practice.

Having read your paper, the patient group is more complex than the title suggests. This patient group is living with severe and complex obesity. Your mean BMI is 63.7kg/m2 at baseline. The biological drivers for individuals with BMI>50kg/m2 are more complex and this should be reflected within the title and manuscript as a novel aspect of this study. Furthermore, the EOSS score suggests that many have additional comorbidities as well.

Finally, there appears to be missing key literature from this paper that is directly relevant to the patient population and the dietary approach used:

Harper C, Maher J, Grunseit A, Seimon RV, Sainsbury A. Experiences of using very low energy diets for weight loss by people with overweight or obesity: a review of qualitative research. Obes Rev. 2018 Oct;19(10):1412-1423.

Astbury NM, Albury C, Nourse R, Jebb SA (2020) Participant experiences of a low-energy total diet replacement programme: A descriptive qualitative study. PLoS ONE 15(9): e0238645.

Christensen B, J, Iepsen E, W, Lundgren J, Holm L, Madsbad S, Holst J, J, Torekov S, S: Instrumentalization of Eating Improves Weight Loss Maintenance in Obesity. Obes Facts 2017;10:633-647.

Please can you consider including these references and how this manuscript adds to the evidence that is already published, this appears key to the framing of this manuscript.

PLease find below some additional points that would benefit from addressing:

Comments:

Line 53: the preferred term is weight-related stigma not obesity-related stigma

Line 62: The term “severely energy-restricted diets” is not a typical term that is used in the literature. Within your review in 2019 you use the terms VLED and LED which are known within the literature and would help the reader to grasp what the manuscript was about. At present I needed to search for what these diets were and read the entire methodology before understanding. Please can you give an energy definition to all the reader to understand.

Line 62: By meal replacements what do you mean? Are you talking about formula total diet replacements that replace the entire diet, partial TDR or meal replacements that replace 1-2 meals/day? This needs clarity as these different dietary approaches give different weight losses and differ in the way they are administered.

Line 77-78: Please consider including qualitative data from other authors here not just Rehackova et al., there is additional insight into the participants experiences which has been missed here. See above for details.

Line 105-119: As a clinician there remains a lack of clarity on what dietary approach you are using when you talk about the SERD, this is important to know what type of diet approach this data relates to. It appears from this description and the reference that you are using a total diet replacement VLED, this needs clarity – also what commercial product did you use.

Line 164-168: It would benefit to see the questions that were asked and how they were framed.

Line 177-183: It is unclear as to why you have used the EOSS tool, what is the relevance to the data being collected at present? As stated above this could be used to identify the complexity of the patient population your studied.

Line 288: Do you have any examples of stigmatising experiences that the participants experienced that made them hesitant

Line 357: There appears to be a feeling of personal failure within this quote was this reflected by others as this can often impact on adherence to dietary programmes with previously held beliefs on dietary success resulting in lack of confidence.

Line 384: Please have consistency with terms regarding stigma suggest using weight-related stigmatising experiences.

Line 387-399: These quotes appear to show feelings of fear related to their medical conditions driving adherence and motivating them to continue e.g. not being able to breathe, fear of death – was there a sense from others that this was an underlying driver?

Line 437: The element of simplicity of diet has been reflected in previous reviews of the topic for the benefit of formula VLED (Brown, A. and Leeds, A.R. (2019), Very low‐energy and low‐energy formula diets: Effects on weight loss, obesity co‐morbidities and type 2 diabetes remission – an update on the evidence for their use in clinical practice. Nutr Bull, 44: 7-24.) and also in qualitative research (Astbury et al., 2020, above). This should be acknowledged that this is already reported and your data supports this view.

Line 438-442: These quotes appear to identify that the use of formula product is the key to the dietary simplicity not just they are choosing from fewer foods.

Line 459-466: Was it the simplicity of the diet that allowed them to plan more easily and get a routine as before this was challenging with all the challenges that food selection requires.

Line 483-487: This infers that all participants had a positive experience from weigh-in and not achieving weight loss, was this universal or did some struggle with this and this impact on adherence to the program?

Line 553: Please change language to person first – ‘disabled participant’ should be ‘participants living with disability’

Line 575: Can you please confirm is this meal replacement product or formula VLED product they are purchasing or did this change as they progressed through the programme?

Line 610: Change terminology related to stigma as before – Also was there a sense of any internalised weight stigma impacted adherence e.g. “I’m lazy” “I’m unmotivated”.

Can I please ask how the question regarding stigma was framed as the quotes appear to suggest that the questions inferred people were feeling stigmatised or did this topic naturally come up in conversation.

Line 708: Is there a possibility that socio-economic factors developed due to the patient sample being mainly socio-economically disadvantaged and that this is simply reflective of your population rather than an issue. Did the issues of cost etc come out of people that were not consider socio-economically disadvantaged?

Line 723: This study offers unique perspective but to an even more unique population that has not been extensively researched before, this is people living with severe and complex obesity e.g. BMI>50kg/m2 and co-morbidities

Line 732: You mention weigh-related stigma being identified in screening do you have a suggestion on how this can be done? Questionnaire? Does this included internalised weight stigma which has been shown to impact healthcare outcomes and motivation to access healthcare. If so then this would benefit from being explore before the conclusion.

Reviewer #2: General Comment:

Overall the study addresses an important issue, which are the facilitators and barriers to patients undertaking severely energy-restricted diets to achieve weight loss.

Introduction:

1. The manuscript describes using severely energy-restricted diets (using meal replacement products) for weight loss. However the authors do not clearly define what is meant by severe energy restriction. Please clarify this.

2. The authors do not provide an adequate background on what is known in the literature about barriers and facilitators to the use of meal replacements (and other interventions using low energy diets) for weight loss and how knowledge gained from their study will add value.

Methods:

3. Much of the information under setting would be better suited as a description of the participants. Please review the subheadings under the methods section.

4. Why did the authors choose to study only participants with class III obesity? Were there any other inclusion/exclusion criteria? The authors described using “purposive sampling”, but did not elaborate on what kind of participants they aimed to include.

5. Duration of interviews and number of in-person/telephone interviews should be reported.

Results:

6. The table of participant characteristics can be organized in a way to help the reader gain a better understanding of the characteristics of participants. Consider arranging according to those who had achieved and maintained weight loss, those who did not achieve weight loss, and dropouts, or any other meaningful arrangement.

7. Please make sure figures in the table are consistently being rounded to the same decimal place.

8. It would be useful for the authors to provide a written summary of the weight loss outcomes achieved by the participants.

Discussion:

9. The authors report that 65% of participants were at socioeconomic disadvantage and this was one of the main barriers identified. How representative is this for the population being studied?

10. The authors do not discuss their findings in context of what is already known in the literature in enough depth.

11. The authors should consider what other limitations there are to the study.

6. PLOS authors have the option to publish the peer review history of their article (what does this mean?). If published, this will include your full peer review and any attached files.

Reviewer #1: No

Reviewer #2: **Yes: **Hadeel Zaghloul

---

## [Author Response · Author response to Decision Letter 0]

24 Mar 2021

Reviewer #1

1. Line 53: the preferred term is weight-related stigma not obesity-related stigma

Our response: Thank you, the manuscript has now been amended accordingly.

2. Line 62: The term “severely energy-restricted diets” is not a typical term that is used in the literature. Within your review in 2019 you use the terms VLED and LED which are known within the literature and would help the reader to grasp what the manuscript was about. At present I needed to search for what these diets were and read the entire methodology before understanding. Please can you give an energy definition to all the reader to understand.

Our response: The term severely energy-restricted diets (SERD) has been used as an umbrella term that includes both VLED and LED protocols, which are used during different stages of the diet program. The diet prescription does not entirely fit the definition of a VLED nor LED, therefore if such terms were used, the paper would be subject to further criticism. Clarity around the term SERD and the link between the terms SERD, VLED and LED has now been provided in the introduction section of the manuscript as seen in the marked-up text. 

3. Line 62: By meal replacements what do you mean? Are you talking about formula total diet replacements that replace the entire diet, partial TDR or meal replacements that replace 1-2 meals/day? This needs clarity as these different dietary approaches give different weight losses and differ in the way they are administered.

Our response: As per the recommendation we have added additional text in the introduction section of the manuscript to provide clarity about the meal replacement diet and products used. The referenced systematic review and meta-analysis and participants in this study used both a VLED and LED total meal replacement diet, and a LED partial meal replacement diet. 

4. Line 77-78: Please consider including qualitative data from other authors here not just Rehackova et al., there is additional insight into the participants experiences which has been missed here. See above for details.

Our response: The listed qualitative data from other authors have now been synthesized into the manuscript in both the introduction (line 108-120) and discussion section (line 890 -895) of the manuscript. 

5. Line 105-119: As a clinician there remains a lack of clarity on what dietary approach you are using when you talk about the SERD, this is important to know what type of diet approach this data relates to. It appears from this description and the reference that you are using a total diet replacement VLED, this needs clarity – also what commercial product did you use.

Our response: Participants in this study used both a VLED total meal replacement diet and a partial meal replacement diet depending on their preference or challenges faced during the course of the program. We have now added additional text to the methods section, subheading “participants”, to provide clarity on the diet used. 

6. Line 164-168: It would benefit to see the questions that were asked and how they were framed.

Our response: Supporting information with the interview guide containing the interview questions have now been submitted to supplemental materials. Text outlining this information has been added to the data collection section. 

7. Line 177-183: It is unclear as to why you have used the EOSS tool, what is the relevance to the data being collected at present? As stated above this could be used to identify the complexity of the patient population your studied.

Our response: The EOSS tool was intended to demonstrate the complexity of the patient population studied. Additional text has been added to provide clarity in the methods section under the subheading “data collection” about the use of the EOSS tool. 

8. Line 288: Do you have any examples of stigmatising experiences that the participants experienced that made them hesitant

Our response: An example of a stigmatising experience has been added by extending the quote in section 1.2 by participant Errol. 

9. Line 357: There appears to be a feeling of personal failure within this quote was this reflected by others as this can often impact on adherence to dietary programmes with previously held beliefs on dietary success resulting in lack of confidence.

Our response: Personal failure was reflected in conversation with one other participant, however it was not the dominant theme. In addition, previous experiences resulting in perceived failings did not seem to affect perceived successful dietary adherence to this dietary program. This is likely because participants considered this program to be different to what they had experienced in the past, largely because of the fast weight loss and support received. Additional text has now been added to section 1.3 to reflect the above.

10. Line 384: Please have consistency with terms regarding stigma suggest using weight-related stigmatising experiences.

Our response: Thank you for bringing this to our attention. The manuscript has been amended to read ‘weight-related stigma’ throughout. 

11. Line 387-399: These quotes appear to show feelings of fear related to their medical conditions driving adherence and motivating them to continue e.g. not being able to breathe, fear of death – was there a sense from others that this was an underlying driver?

Our response: Not all participants were motivated by fear of medical concerns. For example, one participant wanted to travel overseas independently. Another, was motivated by gaining employment in a dream job in the future. Two others with mental health injury recognized that when their mental health deteriorated so did their health and eating patterns. They were able to stick to the diet because of improved mental health, due to other adjunct treatment therapies including psychotherapy. 

The particular quotes in section 1.3 were selected because they related to the sub heading ‘Awareness of eating behaviours and the relationship between these and adverse future health outcomes’. Although the quotes appear to demonstrate fear of worsening medical conditions, they were chosen because it demonstrates intrinsic motivation, following realistic concerns about medical health conditions, which may or may not be fear or anxiety based. 

12. Line 437: The element of simplicity of diet has been reflected in previous reviews of the topic for the benefit of formula VLED (Brown, A. and Leeds, A.R. (2019), Very low‐energy and low‐energy formula diets: Effects on weight loss, obesity co‐morbidities and type 2 diabetes remission – an update on the evidence for their use in clinical practice. Nutr Bull, 44: 7-24.) and also in qualitative research (Astbury et al., 2020, above). This should be acknowledged that this is already reported and your data supports this view.

Our response: We have now included these papers into the 3rd paragraph of the discussion section of the manuscript. 

13. Line 438-442: These quotes appear to identify that the use of formula product is the key to the dietary simplicity not just they are choosing from fewer foods.

Our response: We agree and have added text to section 1.5 to include the observation that it also appears formula products seems to be an important component to the perception of dietary simplicity. 

14. Line 459-466: Was it the simplicity of the diet that allowed them to plan more easily and get a routine, as before this was challenging with all the challenges that food selection requires.

Our response: We agree that the simplicity of the diet and having limited food choice made the diet highly restrictive, and thus made it less cognitively demanding. Participants said simplicity of the diet ensured temptations were not there, and thus implementing a routine was easier, and with a routine compliance with the diet was improved. In addition, the limited foods and repetition of foods from week to week reduced food choice and made food shopping faster and more efficient. Additional text has now been added to section 1.5 (line 592- 594) to reflect the above.

15. Line 483-487: This infers that all participants had a positive experience from weigh-in and not achieving weight loss, was this universal or did some struggle with this and this impact on adherence to the program?

Our response: All participants did not have a positive experience from weigh-ins particularly when what was observed on the scales did not meet their expectations. Lapses in adherence as a result of the weigh-in at the weight loss centre were not apparent during the interviews. Negative feelings seemed to be reconciled by the support provided to participants provided by the facilitator and members of the group session. Participants did acknowledge temporary lapses in adherence when they weighed-in at home, without support. Additional text has been added to section 1.5 paragraph 3, to explain this in more detail. 

16. Line 553: Please change language to person first – ‘disabled participant’ should be ‘participants living with disability’

Our response: The text has been amended to first person language. 

17. Line 575: Can you please confirm is this meal replacement product or formula VLED product they are purchasing or did this change as they progressed through the programme?

Our response: Participants are instructed to purchase a VLED formula meal replacement, and the dietary prescription did change as they progressed through the program. The progression occurred when challenges to adherence were experienced by the participant or they progressed to the next stage of the program at around 12-weeks. The progression included the reduced use of formula meal replacements to using food-based meals in a partial meal replacement diet. The manuscript has been amended to provide this detail in the methods section under the subheading “setting” and in section 2.3, 3rd paragraph as seen in the marked-up text.

18. Line 610: Change terminology related to stigma as before – Also was there a sense of any internalised weight stigma impacted adherence e.g. “I’m lazy” “I’m unmotivated”.

Can I please ask how the question regarding stigma was framed as the quotes appear to suggest that the questions inferred people were feeling stigmatised or did this topic naturally come up in conversation.

Our response: The interviews did not reveal internalized stigma as a factor in poor dietary adherence. All participants, regardless of weight loss outcomes, perceived themselves to be mostly adherent to the diet and motivated, suggesting there was a degree of poor self-awareness in some participants. The actions of turning up to the clinic appointments and participating in the group sessions were perceived by the participant as evidence of their motivation, even when weight loss was not achieved. There was a sense that participants felt misunderstood, because of their body size in that they thought people assumed they were not motivated. This information has been added to section 2.5 as seen in the marked up text. 

The initial question that was asked that evoked a conversation about weight-related stigma was the following: “At any time during the program, did the way you feel about yourself impact on your ability to stick to the meal replacement plan?”

If the participant described a stigmatizing situation, the conversation was then redirected to explore this, by adding an additional question such as; 

“What you are describing is feeling stigmatized, how do you think feeling stigmatized affected your ability to stick to the diet overall?”

The semi-structured interview questions have now been provided in the Interview Guide Supporting Documentation file. Note the above questions were not asked verbatim, but used as a guide and adapted to individual conversations. 

19. Line 708: Is there a possibility that socio-economic factors developed due to the patient sample being mainly socio-economically disadvantaged and that this is simply reflective of your population rather than an issue. Did the issues of cost etc come out of people that were not consider socio-economically disadvantaged?

Our response: The cost of the meal replacement products was a problem for the majority of participants (19 of 23). Of those who did not mention cost, it was typically because they or their partner were employed and receiving a regular income. Being socio-economically disadvantaged affects many things, however, including the ability to live in suitable housing or the social environment in which one lives, even when one is employed. Socio-economic disadvantage appeared to affect program adherence because the situation provided unnecessary disruptions to their life, creating instability and lack of routine. Socio-economic disadvantage is a characteristic of the population in which this sample was chosen. The participants in this study were drawn from a public hospital clinic at which the majority of patients are similarly socio-economically disadvantaged. Furthermore, severe obesity affects those who are more socio-economically disadvantaged to a greater extent than other strata. This information has been added to the discussion section of the manuscript line 822. 

20. Line 723: This study offers unique perspective but to an even more unique population that has not been extensively researched before, this is people living with severe and complex obesity e.g. BMI>50kg/m2 and co-morbidities

Our response: Thank you for this suggestion, we have now emphasized the importance and unique nature of this investigation, as seen in the marked-up text in the last paragraph of the Introduction in the discussion section. 

21. Line 732: You mention weigh-related stigma being identified in screening do you have a suggestion on how this can be done? Questionnaire? Does this included internalised weight stigma which has been shown to impact healthcare outcomes and motivation to access healthcare. If so then this would benefit from being explore before the conclusion.

Our response: 

We were surprised to not find any internalised weight bias in our conversations with patients, however this maybe because it was not specifically asked. We have not made a recommendation to screen for this because it was not a theme found in our research investigation. 

In the discussion section of the manuscript, screening was mentioned for factors that appear to be non-modifiable such as product palatability, socio-economic disadvantage and unforeseeable circumstances. The use of pre-emptive psychoeducation, as a blanket approach, was the intended recommendation made to address weight-related stigma. Upon reviewing these sections, the wording was poor and did not convey the recommendation intended. We have now added text to the discussion section (5th paragraph from the bottom) to provide further clarity to this recommendation. 

Reviewer #2

1. The manuscript describes using severely energy-restricted diets (using meal replacement products) for weight loss. However, the authors do not clearly define what is meant by severe energy restriction. Please clarify this.

Our response: Thank you, the definition of severely energy-restricted diets has now been defined in the 2nd paragraph of the introduction section of the manuscript. 

2. The authors do not provide an adequate background on what is known in the literature about barriers and facilitators to the use of meal replacements (and other interventions using low energy diets) for weight loss and how knowledge gained from their study will add value.

Our response: Thank you, synthesis of existing qualitative research in this topic area has now been added to the 2nd last paragraph of the introduction section. 

3. Much of the information under setting would be better suited as a description of the participants. Please review the subheadings under the methods section.

Our response: Thank you for this recommendation, the subheadings within the section methods have been revised and text reordered, as seen in the marked-up text. 

4. Why did the authors choose to study only participants with class III obesity? Were there any other inclusion/exclusion criteria? The authors described using “purposive sampling”, but did not elaborate on what kind of participants they aimed to include.

Our response: This research study was pragmatic and opportunistic, it was used to explore ways in which clinical practice could be improved for people with class III obesity in a pre-existing service. Participants with class III obesity are a group that is unique and is not well researched, outside of bariatric surgery, thus our interest in shaping clinical practice for this group. The cohort of participants studied was from a pre-existing clinic specifically for individuals with class III obesity, ie patients had to have class III obesity in order to attend this clinic in the first place. Therefore, no additional inclusion/exclusion criteria were needed for this investigation. The method of purposeful sampling and the specific characteristics we were looking for was described in the recruitment section, 2nd paragraph of the manuscript. Additional information to provide clarity on this has been provided in the sections titled Introduction and Methods (subheading “participants”), as seen in the marked-up text.

5. Duration of interviews and number of in-person/telephone interviews should be reported.

Our response: Thank you, this information has been added to the thematic content analysis section, of the manuscript. 

6. The table of participant characteristics can be organized in a way to help the reader gain a better understanding of the characteristics of participants. Consider arranging according to those who had achieved and maintained weight loss, those who did not achieve weight loss, and dropouts, or any other meaningful arrangement.

Our response: Thank you for this suggestion. The table has been reorganised into two groups, those who experienced and maintained >10% weight loss during the time of recruitment and those that did not. 

7. Please make sure figures in the table are consistently being rounded to the same decimal place.

Our response: The figures in the table have now been amended accordingly. 

8. It would be useful for the authors to provide a written summary of the weight loss outcomes achieved by the participants.

Our response: A written summary of the weight loss outcomes have now been provided in the 2nd paragraph of the results section of the manuscript. 

9. The authors report that 65% of participants were at socioeconomic disadvantage and this was one of the main barriers identified. How representative is this for the population being studied?

Our response: Socio-economic disadvantage is a characteristic of the population in which this sample was chosen. The participants in this study were drawn from a public hospital clinic at which the majority of patients are similarly socio-economically disadvantaged. Furthermore, severe obesity affects those who are more socio-economically disadvantaged to a greater extent than other strata. This information has been added to the discussion section of the manuscript. 

10. The authors do not discuss their findings in context of what is already known in the literature in enough depth.

Our response: Further synthesis of the literature has been added to the Introduction (line 108-120) and discussion (line 890 -895) section of the manuscript, as seen in the marked-up text. 

11. The authors should consider what other limitations there are to the study.

Our response: Thank you, additional limitations have now been added to the last paragraph of the discussion section.

---

## [Decision Letter · Decision Letter 1]

27 Apr 2021

PONE-D-20-39574R1

Dietary adherence and program attrition during a severely energy-restricted diet among people with complex class III obesity: a qualitative exploration

PLOS ONE

Dear Dr. Maston,

Thank you for submitting your manuscript to PLOS ONE. After careful consideration, we feel that it has merit but does not fully meet PLOS ONE’s publication criteria as it currently stands. Therefore, we invite you to submit a revised version of the manuscript that addresses the points raised during the review process.

Please address the final reviewer comments.

We look forward to receiving your revised manuscript.

Kind regards,

Shahrad Taheri

Academic Editor

PLOS ONE

Journal Requirements:

Reviewers' comments:

Reviewer's Responses to Questions

**Comments to the Author**

1. If the authors have adequately addressed your comments raised in a previous round of review and you feel that this manuscript is now acceptable for publication, you may indicate that here to bypass the “Comments to the Author” section, enter your conflict of interest statement in the “Confidential to Editor” section, and submit your "Accept" recommendation.

Reviewer #1: (No Response)

Reviewer #2: All comments have been addressed

2. Is the manuscript technically sound, and do the data support the conclusions?

Reviewer #1: Yes

Reviewer #2: Yes

3. Has the statistical analysis been performed appropriately and rigorously? 

Reviewer #1: Yes

Reviewer #2: Yes

4. Have the authors made all data underlying the findings in their manuscript fully available?

Reviewer #1: Yes

Reviewer #2: Yes

5. Is the manuscript presented in an intelligible fashion and written in standard English?

Reviewer #1: Yes

Reviewer #2: Yes

6. Review Comments to the Author

Reviewer #1: Thank you very much for your resubmission of this interesting manuscript on the topic of dietary adherence using severely energy-restricted diet. You have address the suggested comments previously highlighted. There are a couple of very minor addition points from the additional text added:

Line 152: please use the full company names i.e. Cambridge Weight Plan Ltd as the descriptor not Cambridge.

Line 176-180: With the greater explanation of the dietary methods this highlights a question regarding your primary outcome which was dietary adherence - Was there a difference in views between the group that had food and those that chose to just have the meal replacements only? Maybe suggest add a little clarity in the discussion.

Line 729: Your comment regarding “The interviews did not reveal internalized stigma as a factor in poor dietary adherence” is questionable. You did not measure internalised weight stigma using a questionnaire such as Weight bias internalisation scale and did not ask specific questions related to this, therefore how do you know? People living with obesity are often unaware of the internalisation of the weight stigma, as is internalisation of negative social stereotypes such as people with obesity are lazy, gluttonous etc. So it would unlikely come out in the interviews unless specifically addressed. This is more likely to be a limitation of the study and suggest removing this sentence and add to limitations.

Thank you

Reviewer #2: (No Response)

7. PLOS authors have the option to publish the peer review history of their article (what does this mean?). If published, this will include your full peer review and any attached files.

Reviewer #1: No

Reviewer #2: **Yes: **Hadeel Zaghloul

---

## [Author Response · Author response to Decision Letter 1]

11 May 2021

Reviewer #1

1. Line 152: please use the full company names i.e. Cambridge Weight Plan Ltd as the descriptor not Cambridge.

Our response: Thank you, the manuscript has now been amended accordingly starting from line 291. 

2. Line 176-180: With the greater explanation of the dietary methods this highlights a question regarding your primary outcome which was dietary adherence - Was there a difference in views between the group that had food and those that chose to just have the meal replacements only? Maybe suggest add a little clarity in the discussion.

Our response: Participants were specifically asked about the addition of protein and the mode in which it was taken during the SERD, however, it did not illicit a conversation about food-based protein and adherence. Participants who used food-based protein fell into both groups; those that were adherent and those that were not non-adherent. This has now been added to the discussion section of the manuscript, starting at line 1193.

3. Line 729: Your comment regarding “The interviews did not reveal internalized stigma as a factor in poor dietary adherence” is questionable. You did not measure internalised weight stigma using a questionnaire such as Weight bias internalisation scale and did not ask specific questions related to this, therefore how do you know? People living with obesity are often unaware of the internalisation of the weight stigma, as is internalisation of negative social stereotypes such as people with obesity are lazy, gluttonous etc. So, it would unlikely come out in the interviews unless specifically addressed. This is more likely to be a limitation of the study and suggest removing this sentence and add to limitations.

Our response: Thank you for this helpful feedback. Line 729 has now been removed and this limitation has been added the discussion section of the manuscript line 1307.

---

## [Editor Report · Decision Letter 2]

1 Jun 2021

Dietary adherence and program attrition during a severely energy-restricted diet among people with complex class III obesity: a qualitative exploration

PONE-D-20-39574R2

Dear Dr. Maston,

We’re pleased to inform you that your manuscript has been judged scientifically suitable for publication and will be formally accepted for publication once it meets all outstanding technical requirements.

Kind regards,

Shahrad Taheri

Academic Editor

PLOS ONE
---

## [Editor Report · Acceptance letter]

9 Jun 2021

PONE-D-20-39574R2 

Dietary adherence and program attrition during a severely energy-restricted diet among people with complex class III obesity: a qualitative exploration 

Dear Dr. Maston:

I'm pleased to inform you that your manuscript has been deemed suitable for publication in PLOS ONE. Congratulations! Your manuscript is now with our production department. 

Kind regards, 

on behalf of

Dr. Shahrad Taheri 

Academic Editor

PLOS ONE